# School performance in Danish children exposed to maternal type 1 diabetes in utero: A nationwide retrospective cohort study

**Anne Lærke Spangmose**[1‡*], **Niels Skipper**[2,3‡], **Sine Knorr**[4], **Tina Wullum Gundersen**[5], **Rikke Beck Jensen**[6], **Peter Damm**[7,8], **Erik Lykke Mortensen**[9], **Anja Pinborg**[1], **Jannet Svensson**[10,11,12], **Tine Clausen**[8]

**1** Fertility Department, Copenhagen University Hospital, Rigshospitalet, Copenhagen, Denmark, **2** Department of Economics and Business Economics, Aarhus University, Aarhus, Denmark, **3** Centre for Integrated Register-based Research, Aarhus University, Aarhus, Denmark, **4** Steno Diabetes Center Aarhus, Aarhus University Hospital, Aarhus, Denmark, **5** Department of Gynaecology and Obstetrics, Nordsjællands Hospital, Hillerød, Denmark, **6** Department of Growth and Reproduction, Copenhagen University Hospital, Rigshospitalet, Copenhagen, Denmark, **7** Center for Pregnant Women with Diabetes, Department of Obstetrics, Copenhagen University Hospital, Rigshospitalet, Copenhagen, Denmark, **8** Department of Clinical Medicine, University of Copenhagen, Denmark, **9** Department of Public Health and Center for Healthy Aging, University of Copenhagen, Denmark, **10** Department of Paediatric and Adolescents, Copenhagen University Hospital, Herlev-Gentofte, Denmark, **11** Department of Clinical Medicine, University of Copenhagen, Copenhagen, Denmark, **12** Steno Diabetes Center Copenhagen, Copenhagen, Denmark

‡ These authors share first authorship on this work.
* anne.laerke.spangmose.pedersen@regionh.dk

**Data Availability Statement:** The data are drawn from a restricted access data source and cannot be

## Abstract

### Background

Conflicting results have been reported concerning possible adverse effects on the cognitive function of offspring of mothers with type 1 diabetes (O-mT1D). Previous studies have included offspring of parents from the background population (O-BP), but not offspring of fathers with type 1 diabetes (O-fT1D) as the unexposed reference group.

### Methods and findings

This is a population-based retrospective cohort study from 2010 to 2016. Nationally standardized school test scores (range, 1 to 100) were obtained for public school grades 2, 3, 4, 6, and 8 in O-mT1D and compared with those in O-fT1D and O-BP. Of the 622,073 included children, 2,144 were O-mT1D, and 3,474 were O-fT1D. Multiple linear regression models were used to compare outcomes, including the covariates offspring with type 1 diabetes, parity, number of siblings, offspring sex, smoking during pregnancy, parental age, and socioeconomic factors. Mean test scores were 54.2 (standard deviation, SD 24.8) in O-mT1D, 54.4 (SD 24.8) in O-fT1D, and 56.4 (SD 24.7) in O-BP. In adjusted analyses, the mean differences in test scores were −1.59 (95% CI −2.48 to −0.71, $p < 0.001$) between O-mT1D and O-BP and −0.78 (95% CI −1.48 to −0.08, $p = 0.03$) between O-fT1D and O-BP. No significant difference in the adjusted mean test scores was found between O-mT1D and O-fT1D ($p = 0.16$). The study's limitation was no access to measures of glycemic control during pregnancy.

made publicly available. The data used in the paper are administrative Danish data maintained by Statistics Denmark, which is kept in a secure server. However, the data can be accessed remotely from within Danish universities and research institutions. If a researcher at a university or other research institution outside Denmark wishes to use these data, this may be accomplished by visiting a Danish research institution or cooperating with researchers or research assistants working in Denmark. For inquiries, visit www.dst.dk or e-mail: dst@dst.dk.

**Funding:** ALS received funding from The Research Fund of Rigshospitalet, Copenhagen University Hospital; NS received funding from Det Frie Forskningsråd (Award number: 8019-00055B). The funders had no role in study design, data collection and analysis, decision to publish, or preparation of the manuscript.

**Competing interests:** I have read the journal's policy and the authors of this manuscript have the following competing interests: all authors have completed the ICMJE uniform disclosure form at www.icmje.org/coi_disclosure.pdf and declare no support from any organization for the submitted work; JS reported serving as an adviser to Medtronic and Novo Nordisk, to owning shares in Novo Nordisk, and to receiving speaking fees from Medtronic and Novo Nordisk; PD participate in clinical studies in pregnant women in collaboration with Novo Nordisk, no personal honorarium is involved; no other relationships or activities that could appear to have influenced the submitted work.

**Abbreviations:** aOR, adjusted odds ratio; ICD, International Classification of Diseases; IGF1, insulin-like growth factor 1; O-BP, offspring of parents from the background population; O-fT1D, offspring of fathers with type 1 diabetes; O-mT1D, offspring of mothers with type 1 diabetes; OR, odds ratio; SD, standard deviation; STROBE, Strengthening the Reporting of Observational Studies in Epidemiology.

## Conclusions

O-mT1D achieved lower test scores than O-BP but similar test scores compared with O-fT1D. Glycemic control during pregnancy is essential to prevent various adverse pregnancy outcomes in women with type 1 diabetes. However, the present study reduces previous concerns regarding adverse effects of in utero hyperglycemia on offspring cognitive function.

## Author summary

### Why was this study done?

- The influence of maternal diabetes during pregnancy on offspring cognition has been widely explored because high blood sugar levels in pregnant women are suspected to affect fetal development, including the brain.

- Limited evidence is available on how different subtypes of maternal diabetes are associated with offspring cognition (e.g., gestational diabetes and type 1 and type 2 diabetes).

- Limited data are available on the association between maternal type 1 diabetes during pregnancy and offspring cognition using offspring of fathers with type 1 diabetes (O-fT1D) as the unexposed reference group, which allows for adjustments of potentially shared genes and familial stress of having a parent suffering from a serious chronic disease like diabetes.

### What did the researchers do and find?

- Data were identified from Danish registers, and test scores in math (grades 3 and 6) and reading (grades 2, 4, 6, and 8) were obtained on all Danish children attending public schools from 2010 to 2016.

- We included 2,144 offspring of mothers and 3,474 O-fT1D and 616,455 from the background population, including 1,704,447 test scores.

- Offspring of mothers with type 1 diabetes (O-mT1D) achieved lower test scores than offspring in the background population but similar test scores compared with O-fT1D.

### What do these findings mean?

- The lower test scores in the O-mT1D appear to reflect a negative association of having a parent with type 1 diabetes rather than a specific adverse effect of maternal type 1 diabetes during pregnancy on the fetus.

- This study presents evidence of an alternative explanation for the previously observed adverse effect of maternal type 1 diabetes during pregnancy on offspring cognitive development.

## Introduction

The influence of maternal diabetes during pregnancy on offspring cognition has been widely explored [1–10]. Unlike insulin, glucose crosses the placenta, and, therefore, maternal hyperglycemia leads to intrauterine hyperglycemia and subsequently fetal hyperinsulinemia, which increases fetal growth [11]. In addition, maternal hyperglycemia serves as a nonspecific teratogenic factor. In animal studies, impaired dendritic development is described in offspring born to rats with streptozotocin-induced hyperglycemia in pregnancy—possibly through abnormal insulin/insulin-like growth factor 1 (IGF1) signaling in the fetal brain [10]. Other studies have found an up-regulation of IGF-1 receptors in the cerebellum and structural changes in the hypothalamus in the offspring of diabetic rats [12,13]. However, few human studies on cognitive function in offspring exposed to maternal diabetes during pregnancy have distinguished between different types of maternal diabetes (e.g., gestational diabetes and type 1 and type 2 diabetes). These studies were based on small sample sizes and a wide variety of cognitive tests, and they reached varying conclusions [1–6]. Furthermore, to our knowledge, previous studies have included offspring of parents from the background population (O-BP), but not offspring of fathers with type 1 diabetes (O-fT1D) as the unexposed reference group.

All Danish children in public schools are tested in reading and math using nationally standardized tests. This provides a unique opportunity to assess the potential association between in utero exposure to maternal type 1 diabetes and school performance in offspring of mothers with type 1 diabetes (O-mT1D). Therefore, the primary objective of this study was to compare test scores in O-mT1D with test scores in O-fT1D and O-BP. We hypothesized that O-mT1D would achieve lower test scores compared with O-fT1D and O-BP due to intrauterine exposure of hyperglycemia.

## Methods

This population-based retrospective cohort study included all singletons attending public schools in Denmark from January 1, 2010 to December 31, 2016 (S1 Fig). The analysis plan was drafted prospectively in June 2020 (S1 Text). This study is reported as per the Strengthening the Reporting of Observational Studies in Epidemiology (STROBE) guideline (S1 Checklist).

Offspring were initially identified through the national Medical Birth Register to ensure inclusion of offspring born in Denmark in whom there was valid information on maternal and paternal diabetes diagnosed before pregnancy. Subsequently, maternal, obstetric, and perinatal outcomes from the Medical Birth Register were cross-linked to in- and outpatient diagnoses from the National Patient Register and cross-linked to nationally standardized school test scores and socioeconomic factors from different administrative registers from Statistics Denmark. This cross-linking of registers is possible due to the unique personal identification number allocated to all residents in Denmark either at birth or on immigration. All data were deidentified prior to access.

### Exposure

Type 1 diabetes was defined according to the International Classification of Diseases (ICD) 8th revision (ICD-8) 249 code and the 10th revision (ICD-10) E10 code assigned before childbirth.

Offspring were grouped by parental diabetes status: O-mT1D, O-fT1D, or O-BP. Offspring in whom both the mother and the father were diagnosed with type 1 diabetes were excluded (S1 Fig).

## Outcomes

During the follow-up period, nationally standardized school tests were mandatory in Danish public schools, but not private schools. Consequently, outcomes based on test scores were restricted to offspring attending public schools.

The children were tested in math in grades 3 and 6 and in reading in grades 2, 4, 6, and 8. The tests evaluate the students' abilities in math and reading within different profile areas. In math, the profile areas include algebra, geometry, statistics, and probability, while the reading profile areas include language and text comprehension and decoding. The tests are computer based, and the students are presented with questions of varying difficulty to assess their individual level of skills. At the end of the test, a score is automatically calculated, ranging from 1 to 100 [14,15]. The scores from these tests are highly correlated with the grade point average on compulsory school exit exams, and the tests have been used for research in several previous studies [14,16–18].

The primary outcome was the test scores in both math and reading. Secondary outcomes were test scores analyzed separately in math or reading and test scores in math or reading stratified by grade. Third, the likelihood of missing a test despite attending a public school during follow-up was assessed. Finally, the likelihood of attending a private school was assessed to check if the studied population was representative.

## Covariates

Covariates were included to compare outcomes across grades (categorical: 2, 3, 4, 6, or 8), topic (math/reading), and year of test (categorical: 2010, 2011, 2012, 2013, 2014, 2015, or 2016).

Basic covariates were offspring sex (male/female), parity (categorical: 1, 2, 3, 4, or $\geq$5), number of siblings (categorical: 0, 1, 2, or $\geq$3), offspring with type 1 diabetes (yes/no) defined as ICD-10 E10 code assigned before a test, and maternal smoking during pregnancy (yes/no).

Parental socioeconomic factors were parental highest educational level (categorical: primary school, high school, vocational, short higher education, middle secondary education, master's degree, or higher), income (categorical: percentile rank Q1, Q2, Q3, or Q4), immigrant or descendent status (yes/no), and age (continuous). Furthermore, parents living together (yes/no) was included as a socioeconomic variable. All socioeconomic factors in the offspring were recorded at age 5 (before starting school).

Obstetric and perinatal covariates acting as potential mediators were hypertensive disorders of pregnancy (yes/no) defined as the ICD-10 codes O10 to 16 assigned during the 280 days before date of birth, cesarean section (yes/no), low Apgar score at 5 minutes ($<7$ or $\geq7$), gestational age (categorical: $<32 + 0$, $32 + 0$ to $36 + 6$, $37 + 0$ to $39 + 6$, or $\geq40 + 0$ weeks), and birth weight according to expected sex-specific birth weight for the given gestational age, calculated according to Marsal and colleagues (categorical: small, average, or large for gestational age) [19]. Small, average, and large for gestational age were defined as the $<10\%$, $10\%$ to $90\%$, and $>90\%$ percentile, respectively.

## Statistical analyses

Descriptive statistics regarding maternal, paternal, and offspring characteristics were presented as proportions or means with standard deviations (SDs; Table 1).

The association between O-mT1D, O-fT1D, and O-BP and test scores was estimated in multiple regression models with O-BP as reference. As a child could take multiple tests, hence, contributing with multiple test scores (math or reading, but also at different grade levels), test scores were analyzed in a pooled analysis, with each test score as a unique observation and

**Table 1. Descriptive statistics of study sample.**

| Characteristics | % O-mT1D (n = 2,144) | % missing obs. | O-fT1D (n = 3,474) | % missing obs. | O-BP (n = 616,455) | % missing obs. | F-test for no differences across groups |
|---|---|---|---|---|---|---|---|
| **Child** | | | | | | | |
| Male | 52.7 | 0% | 50.2 | 0% | 51.2 | 0% | 0.21 |
| Female | 47.3 | 0% | 49.8 | 0% | 48.7 | 0% | 0.20 |
| Firstborn | 44.9 | 0% | 42.4 | 0% | 43.2 | 0% | 0.18 |
| Number of siblings, mean (SD) | 1.2 (0.78) | 0% | 1.4 (0.82) | 0% | 1.5 (0.8) | 0% | <0.001 |
| Parents living together | 64.5 | 0% | 61.5 | 0% | 67.3 | 0% | <0.001 |
| **Mother** | | | | | | 1% | <0.001 |
| Age, mean (SD) | 35.5 (4.67) | 0% | 35.2 (4.83) | 0% | 35.0 (4.75) | 0% | <0.001 |
| Master's degree or higher | 6.0 | 1% | 7.0 | 1% | 9.1 | 1% | <0.001 |
| Income percentile rank, mean (SD) | 0.60 (0.23) | 1% | 0.63 (0.22) | 1% | 0.6 (0.22) | 1% | <0.001 |
| Immigrant or descendant[a] | 6.4 | 2% | 10.8 | 3% | 11.3 | 2% | <0.001 |
| **Father** | | | | | | 0% | <0.001 |
| Age, mean (SD) | 38.0 (5.67) | 2% | 38.3 (6.01) | 3% | 37.6 (5.62) | 2% | <0.001 |
| Master's degree or higher | 8.2 | 2% | 7.7 | 3% | 10.8 | 2% | <0.001 |
| Income percentile rank, mean (SD) | 0.74 (0.24) | 0% | 0.7 (0.26) | 0% | 0.7 (0.24) | 0% | <0.001 |
| Immigrant or descendant[a] | 7.8 | 2% | 9.5 | 3% | 11.5 | 2% | <0.001 |
| **Clinical** | | | | | | | |
| Offspring with type 1 diabetes | 2.1 | 0% | 4.0 | 0% | 0.4 | 0% | 0.03 |
| Maternal smoking during pregnancy | 30.5 | 16% | 26.9 | 12% | 28.2 | 14% | 0.03 |
| Hypertensive disorders of pregnancy | 16.8 | 0% | 4.5 | 0% | 4.2 | 0% | <0.001 |
| Cesarean section | 59.6 | 0% | 16.5 | 6% | 16.1 | 0% | <0.001 |
| Low Apgar score at 5 minutes (<7) | 1.2 | 2% | 0.6 | 1% | 0.6 | 1% | 0.006 |
| Gestational age <37+0 | 31.5 | 10% | 4.2 | 9% | 4.4 | 11% | <0.001 |
| Large for gestational age | 59.2 | 10% | 13.6 | 9% | 13.5 | 11% | <0.001 |
| **Test** | | | | | | | |
| Math and reading test score, mean (SD) (n = 1,704,447) | 54.2 (24.8) | | 54.4 (24.8) | | 56.4 (24.7) | | <0.001 |
| Math test score, mean (SD) (n = 574,160 test scores) | 53.2 (25.1) | | 53.9 (25.2) | | 56.5 (25.1) | | <0.001 |
| Reading test score, mean (SD) (n = 1,130,287 test scores) | 54.7 (24.7) | | 54.7 (24.5) | | 56.4 (24.5) | | <0.001 |
| Missing test scores | 5.5 | | 4.4 | | 4.1 | | <0.001 |

Notes: Data reported as proportions or means with SD in parentheses. *p*-Value from F-test for no differences in means across the groups O-mT1D (maternal diabetes), O-fT1D (paternal diabetes), and O-BP (background population).

[a]Immigrants are Danish residents who were not born in Denmark and had patents who were not born in Denmark. Descendants are Danish residents who were born in Denmark and had parents who were not born in Denmark.

O-BP, offspring of parents from the background population; O-fT1D, offspring of fathers with type 1 diabetes; O-mT1D, offspring of mothers with type 1 diabetes; SD, standard deviation.

cluster-robust standard errors were used to adjust for within-individual correlations. To accommodate clustering of data at the school level, a school fixed effect regression model was estimated. To assess whether test scores in O-mT1D were significantly different from those in O-fT1D, we performed a Wald test for the equality of coefficients in the 2 groups in the regression model. The mean difference in test scores was estimates in a linear regression model. Selected

mean differences in test scores were also presented as Cohen's *d* to assess the magnitude of the differences. The risk of missing a test score and the odds ratio (OR) of attending a private school were estimated as an OR in logistic regression models. Covariates were chosen based on current knowledge about factors associated with intelligence and school performance [14,20].

Crude analyses of the pooled test scores in math and reading included only adjustment for grade, topic (reading/math), and year effects (grade–topic–year–specific effects) (Model 1). In Model 2, the multiple regression was further adjusted for the following basic covariates: offspring sex, parity, number of siblings, offspring with type 1 diabetes, and maternal smoking during pregnancy. Model 3 was further adjusted for socioeconomic factors including parental age. Finally, to evaluate a potential mediating role of obstetric and perinatal complications, these covariates were added to Model 3 one by one in separate analyses (Models S1–S5) and in a combined analysis (Model S6).

In multiple regression analyses that included the covariates in Model 3, we estimated the OR of attending a private school (logistic regression analyses), the mean differences in test scores stratified separately by math or reading and test scores in math or reading stratified by grade (linear regression analyses), and the OR of missing a test score (logistic regression analyses). To investigate the importance of missing data on test scores conditional on being enrolled in a public school, 2 different sensitivity analyses were performed on the main outcome: (1) all children with missing test scores were assigned the lowest possible score, and the main analysis was reestimated to see whether conclusions changed; and (2) inverse probability weighting was undertaken using the predictions from the logistic regression of missing a test score to reweight for nonresponse in the main analysis [21].

Missing values on independent variables in the regression models were addressed by adding dummy variables for missing values (Table 1).

A *p*-value <0.05 was considered statistically significant. Because of the potential for type I errors due to multiple comparisons, results for analyses of secondary outcomes should be interpreted as exploratory. All statistical analyses were performed using Stata 15 (Stata).

## Ethical approval

The study was approved by the Danish Data Protection Agency. This approval constitutes the necessary legal requirement, and informed consent is not required.

## Results

Of 731,455 children initially identified via Statistics Denmark, 622,073 attended public schools (85%), and 1,704,447 test scores were obtained. Among children in public schools, 2,144 O-mT1D, 3,474 O-fT1D, and 616,455 O-BP were eligible for the study (S1 Fig). The adjusted odds ratios (aORs) for attending a private school were 1.02 (95% CI 0.89 to 1.17, *p* = 0.85) for O-mT1D versus O-BP and 0.94 (95% CI 0.84 to 1.05, *p* = 0.17) for O-fT1D versus O-BP. There were no significant different in the aOR of attending a private school for O-mT1D versus O-fT1D (*p* = 0.31). Among the offspring attending 1 or more grades, 96% O-mT1D, 97% O-fT1D, and 97% O-BP were registered with at least 1 test score. More O-mT1D had at some point missed a test score than O-BP (aOR 1.30 [95% CI 1.16 to 1.47, *p* < 0.001]), but no difference was observed for O-fT1D versus O-BP (aOR 1.02 [95% CI 0.92 to 1.13, *p* = 0.80]). O-mT1D were significantly more likely to have a missing test score compared with O-fT1d (*p* = 0.003).

In total, 2,144 offspring of 1,597 unique mothers and 3,474 offspring of 2,419 unique fathers with type 1 diabetes and 616,455 offspring of 401,406 unique mothers and 402,785 unique fathers from the background population had offspring included in the cohort. Offspring of both O-mT1D and O-fT1D were excluded (*n* = 30).

## Characteristics of the population

The mean age of the study population was 10.1 years (SD 2.39) (at first test). Of the study population, 98.3% were born between 1996 and 2007. Comparisons of background characteristics showed that O-mT1D had fewer siblings, their mothers were older and less likely to have a master's degree or higher educational level, and their parents were less likely to be immigrants or descendants than O-BP and O-fT1D. Obstetric and perinatal complications were more often present in pregnancies in O-mT1D.

Offspring living with both parents accounted for 64.5% in O-mT1D, 61.5% in O-fT1D, and 67.3% in O-BP. Mean paternal age was higher in O-mT1D, and their fathers were less likely to have a master's degree or higher educational level than O-BP. Mean paternal age was lower in O-mT1D, and their fathers were more likely to have a master's degree or higher educational level than O-fT1D. Diabetes was diagnosed in 2.1% of O-mT1D, 4.0% of O-fT1D, and 0.4% of O-BP (Table 1).

## Test scores

The mean of test scores were 54.2 (SD 24.8) in O-mT1D, 54.4 (SD 24.8) in O-fT1D, and 56.4 (SD 24.7) in O-BP, corresponding to a mean difference of −1.90 [95% CI −2.87 to −0.94, $p < 0.001$] (Cohen's $d$ = −0.08) for O-mT1D versus O-BP and −1.57 [95% CI −2.33 to −0.81, $p < 0.001$] (Cohen's $d$ = −0.06) for O-fT1D versus O-BP after adjusting for the grade–topic–year–specific effects (Table 2, Model 1). After adjusting for basic and socioeconomic covariates, O-mT1D and O-fT1D had lower mean test scores than O-BP (mean differences −1.59 [95% CI −2.48 to −0.71, $p < 0.001$] (Cohen's $d$ = −0.06) and −0.78 [95% CI −1.48 to −0.08, $p = 0.03$] (Cohen's $d$ = −0.03), respectively) (Table 2, Model 3). The mean difference in test score for O-mT1D versus O-fT1D was insignificant in both Model 1 and 3 ($p = 0.60$ and $p = 0.16$, respectively) (Table 2). Analyses that included further adjustments for all potentially mediating obstetric and perinatal covariates did not change the results (S1 Table, Model S1–S6).

In the first sensitivity analysis assessing the potential significance of the increased probability of missing a test score among O-mT1D and O-fT1D, the main analysis was estimated (Model 3) after recoding all missing test scores to 1 (assuming those missing the test scores would have performed worst of all). The mean adjusted test scores differences were −1.73 [95% CI −2.64 to −0.82, $p < 0.001$] for O-mT1D versus O-BP and −0.82 [95% CI −1.54 to −0.10, $p = 0.03$] for O-fT1D versus O-BP. The difference between O-mT1D and O-fT1D was insignificant ($p = 0.12$).

In the second sensitivity analysis using inverse probability weighting for nonresponse, the mean adjusted (Model 3) test score differences were −1.60 [95% CI −2.49 to −0.71, $p < 0.001$] for O-mT1D versus O-BP and −0.71 [95% CI −1.42 to −0.00, $p = 0.049$] for O-fT1D versus O-BP. The difference between O-mT1D and O-fT1D was not statistically significant ($p = 0.13$).

## Test scores in math or reading

In adjusted analyses that included covariates in Model 3, the overall mean differences in math test scores were lower for both O-mT1D and O-fT1D compared with O-BP (mean difference −2.61 [95% CI −3.71 to −1.50, $p < 0.001$] and −1.19 [95% CI −2.08 to −0.31, $p = 0.01$], respectively). Similarly, analyses of math test scores stratified by grades showed that both O-mT1D and O-fT1D had lower mean test scores in third grade compared with O-BP (mean difference −2.38 [95% CI −3.82 to −0.94, $p = 0.001$ and −1.17 [95% CI −2.27 to −0.07, $p = 0.04$], respectively). However, in sixth grade, only O-mT1D were found to have a lower mean test score than O-BP (mean difference −2.77 [95% CI −4.17 to −1.40, $p < 0.001$]) (Fig 1).

**Table 2. Multiple linear regression analyses comparing test scores in offspring of mothers (O-mT1D) and fathers (O-fT1D) with type 1 diabetes compared with O-BP, mean test score difference (95% CI).**

| Explanatory variables | Model ($n$ = 1,704,447 test scores) | | |
| --- | --- | --- | --- |
| | Model 1 | Model 2 | Model 3 |
| | Mean difference (95% CI), $p$-value | Mean difference (95% CI), $p$-value | Mean difference (95% CI), $p$-value |
| **Diabetes status** | | | |
| O-mT1D | −1.90 (−2.87 to −0.94), $p < 0.001$ | −1.96 (−2.91 to −1.01), $p < 0.001$ | −1.59 (−2.48 to −0.71), $p < 0.001$ |
| O-fT1D | −1.57 (−2.33 to −0.81), $p < 0.001$ | −1.49 (−2.23 to −0.74), $p < 0.001$ | −0.78 (−1.48 to −0.08), $p = 0.03$ |
| O-BP | (ref.) | (ref.) | (ref.) |
| **Basic covariates** | | | |
| Male | | −2.88 (−2.99 to −2.77), $p < 0.001$ | −2.98 (−3.08 to −2.87), $p < 0.001$ |
| Parity | | | |
| 1 | | (ref.) | (ref.) |
| 2 | | −2.67 (−2.80 to −2.54), $p < 0.001$ | −2.96 (−3.09 to −2.84), $p < 0.001$ |
| 3 | | −3.96 (−4.16 to −3.76), $p < 0.001$ | −4.65 (−4.85 to −4.45), $p < 0.001$ |
| 4 | | −4.20 (−4.59 to −3.80), $p < 0.001$ | −5.46 (−5.85 to −5.08), $p < 0.001$ |
| ≥5 | | −9.02 (−9.58 to −8.46), $p < 0.001$ | −8.00 (−8.55 to −7.44), $p < 0.001$ |
| Number of siblings | | | |
| 0 | | (ref.) | (ref.) |
| 1 | | 1.95 (1.73 to 2.18), $p < 0.001$ | 0.51 (0.29 to 0.73), $p < 0.001$ |
| 2 | | 1.85 (1.61 to 2.10), $p < 0.001$ | 0.68 (0.44 to 0.92), $p < 0.001$ |
| ≥3 | | −2.19 (−2.51 to −1.88), $p < 0.001$ | −0.22 (−0.52 to 0.09), $p = 0.17$ |
| Offspring with type 1 diabetes | | −0.09 (−0.97 to 0.80), $p = 0.84$ | −0.16 (−0.98 to 0.67), $p = 0.71$ |
| Maternal smoking during pregnancy | | −6.29 (−6.44 to −6.14), $p < 0.001$ | −2.08 (−2.23 to −1.93), $p < 0.001$ |
| **Socioeconomic covariates** | | | |
| Mother's highest educational level | | | |
| Primary school | | | (ref.) |
| High school | | | 7.36 (7.12 to 7.60), $p < 0.001$ |
| Vocational | | | 3.27 (3.09 to 3.44), $p < 0.001$ |
| Short higher education | | | 7.1 (6.82 to 7.38), $p < 0.001$ |
| Middle secondary education | | | 8.75 (8.55 to 8.94), $p < 0.001$ |
| ≥Master's degree | | | 12.59 (12.34 to 12.84), $p < 0.001$ |
| Mother's income, by quartile in income distribution | | | |
| Q1 | | | (ref.) |
| Q2 | | | 0.13 (−0.17 to 0.43), $p = 0.40$ |
| Q3 | | | 0.84 (0.67 to 1.02), $p < 0.001$ |
| Q4 | | | 2.27 (2.07 to 2.46), $p < 0.001$ |
| Mother is immigrant or descendant* | | | −3.42 (−3.67 to −3.18), $p < 0.001$ |
| Father's highest educational level | | | |
| Primary school | | | (ref.) |
| High school | | | 7.87 (7.61 to 8.12), $p < 0.001$ |
| Vocational | | | 2.76 (2.60 to 2.91), $p < 0.001$ |
| Short higher education | | | 6.29 (6.05 to 6.54), $p < 0.001$ |
| Middle secondary education | | | 8.9 (8.69 to 9.10), $p < 0.001$ |
| ≥Master's degree | | | 11.35 (11.12 to 11.58), $p < 0.001$ |
| Father's income, by quartile in income distribution | | | |
| Q1 | | | (ref.) |
| Q2 | | | 0.12 (−0.22 to 0.46), $p = 0.49$ |
| Q3 | | | −0.03 (−0.23 to 0.17), $p = 0.74$ |

(*Continued*)

**Table 2.** (Continued)

| Explanatory variables | Model ($n$ = 1,704,447 test scores) | | |
| --- | --- | --- | --- |
| | Model 1 | Model 2 | Model 3 |
| | Mean difference (95% CI), $p$-value | Mean difference (95% CI), $p$-value | Mean difference (95% CI), $p$-value |
| Q4 | | | 1.18 (1.00 to 1.36), $p < 0.001$ |
| Father is immigrant or descendant* | | | −2.96 (−3.21 to −2.72), $p < 0.001$ |
| Parents living together | | | 1.73 (1.61 to 1.85), $p < 0.001$ |
| **Test: ($p$-value)** | | | |
| O-mT1D = O-fT1D | 0.60 | 0.44 | 0.16 |

Notes: All regression models are adjusted for grade-, topic-, and year-specific fixed effects.

Model 1 is with no further adjustment.

Model 2 is adjusted for offspring sex, parity, number of siblings, offspring with type 1 diabetes, and maternal smoking during pregnancy

Model 3 is adjusted for offspring sex, parity, number of siblings, offspring with type 1 diabetes, and maternal smoking during pregnancy, parental highest educational level, income, immigrant or descendant status, age (coefficients not reported), and parents living together.

All covariates are dichotomous (0/1).

$p$-Value from Wald test (F-test) of equality of the regression coefficients to O-mT1D (maternal diabetes) and O-fT1D (paternal diabetes) is reported.

*Immigrants are Danish residents not born in Denmark, with neither of their parents born in Denmark. Descendants are Danish residents born in Denmark, with neither of their parents born in Denmark.

O-BP, offspring of parents from the background population; O-fT1D, offspring of fathers with type 1 diabetes; O-mT1D, offspring of mothers with type 1 diabetes.

O-mT1D were found to have a lower overall reading test score (mean difference −1.10 [95% CI −2.05 to −0.14, $p$ = 0.02]) and lower reading test scores in fourth grade than O-BP (mean difference −2.19 [95% CI −3.58 to −0.80, $p$ = 0.002]) (Fig 1).

Neither math test scores nor reading test scores differed for O-mT1D versus O-fT1D except for the overall math test score, which were lower for O-mT1D ($p$ = 0.0496) (Fig 1).

## Discussion

### Principal findings

In this retrospective register-based cohort study, we found that both O-mT1D and O-fT1D had lower test scores than O-BP. However, there were no differences in test scores for O-mT1D and O-fT1D. O-mT1D had a slightly increased risk of missing a test score compared with O-fT1D and a lower overall math test score.

### Comparisons to other studies

Several previous studies have examined the association between exposure to maternal diabetes during pregnancy and cognitive ability in offspring [1–10]. However, to our knowledge, only 4 existing studies have distinguished between type 1 diabetes, type 2 diabetes, and gestational diabetes [1–4], 3 of which are small clinical follow-up studies [1,2,4]. Similar, we are not aware of previous studies comparing O-mT1D with O-fT1D, hence taking advantage of the ability to adjust for potential genetic factors. The 4 previous studies of O-mT1D addressed cognitive function in different ways; some estimated intelligence quotients using standardized tests [1,2,4], while one study used mandatory school exit exams as a proxy for cognitive function [3]. One study found O-mT1D to have impaired cognitive function compared with O-BP [4], while 3 studies found similar cognitive function [1–3]. Two of the studies were included in a recent systematic review and meta-analysis, concluding that exposure to maternal type 1 diabetes during pregnancy is associated with a lower intelligence quotient in the offspring (pooled weighted mean difference −4.62 (95% CI −6.75 to −2.50) [1,4,5]. Among these 4 studies, the

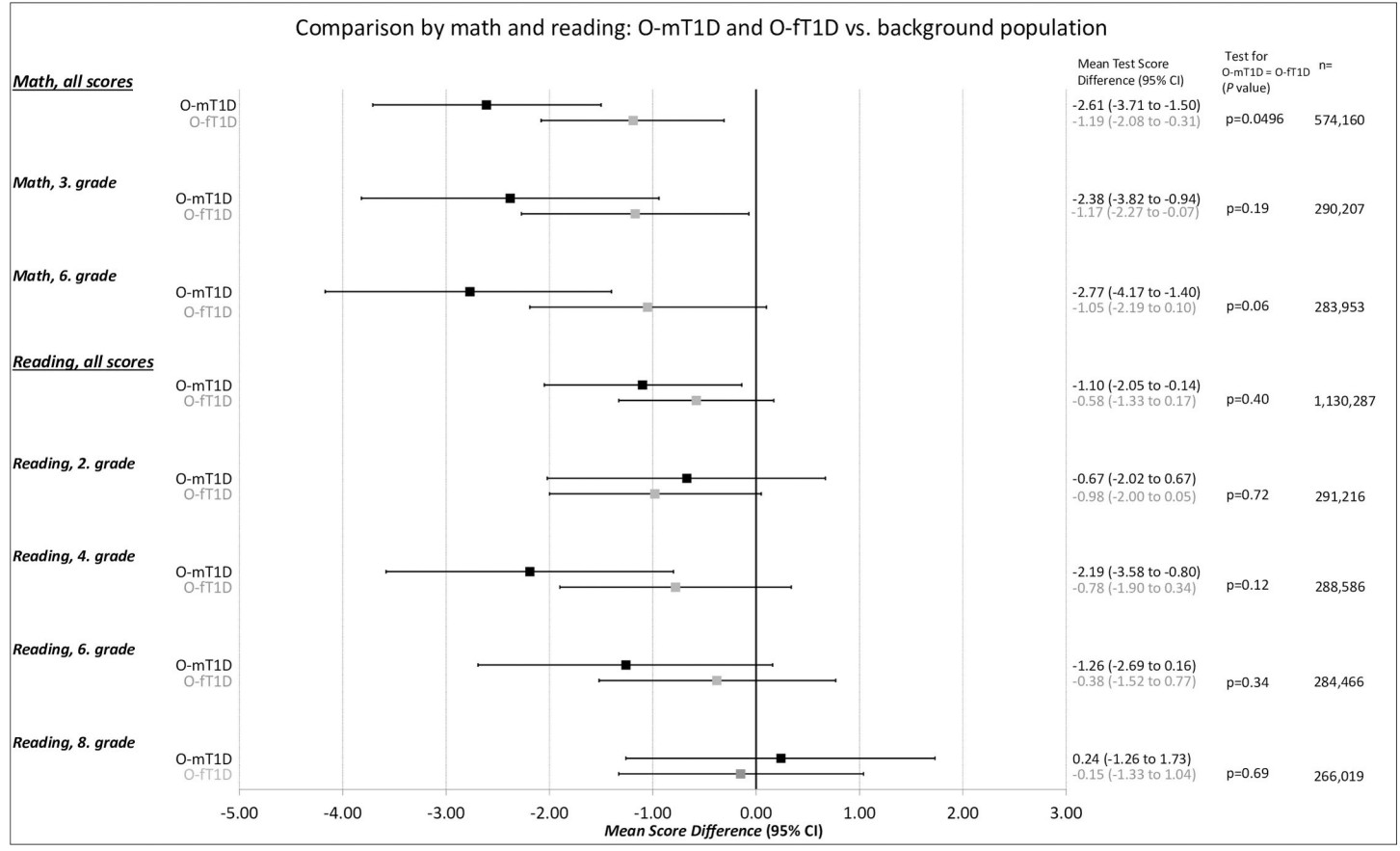

**Fig 1. Multiple linear regression analyses comparing test scores in math or reading in offspring of mothers (O-mT1D) and fathers (O-fT1D) with type 1 diabetes compared with O-BP, mean test score difference (95% CI).** Notes: Differences are adjusted for grade-, topic-, and year-specific fixed effects, offspring sex, parity, number of siblings, offspring with type 1 diabetes, maternal smoking during pregnancy, parental highest educational level, income, immigrant or descendant status, age, and parents living together (corresponding to Model 3). *p*-Value from Wald test (F-test) of equality of the regression coefficients to O-mT1D and O-fT1D is reported. O-BP, offspring of parents from the background population; O-fT1D, offspring of fathers with type 1 diabetes; O-mT1D, offspring of mothers with type 1 diabetes.

study by Knorr and colleagues had the design that was most similar to our study. They assessed mandatory school exit exam grades, which are shown to be highly correlated with the test scores used in our study [14,16]. Knorr and colleagues included O-mT1D from a prospectively sampled clinical cohort (*n* = 707) born in Denmark from 1992 to 1999 and found that O-mT1D had an insignificant, slightly lower adjusted mean grade point average than O-BP (adjusted mean difference –0.07 [95% CI –0.23 to 0.09]) [3]. Our study included test scores from multiple grades and a cohort of O-mT1D that was 3-fold larger than in the study by Knorr and colleagues, which possibly explains why our results reached statistical significance contrary to the study by Knorr and colleagues Moreover, the outcome of the current study was a low stake, computerized, and nonteacher-assessed test. The differences in test scores between O-mT1D and O-BP in our study, represented as Cohen's *d*, were not larger than 0.1 (numerically), and a Cohen's *d* less than 0.20 is generally considered a small effect size [22].

## Strengths and limitations

The major strength of this study is the population design. The register-based study design enabled us to include all offspring attending public schools over a 7-year period and allowed us to correct for covariates shown to be strongly associated with intelligence and school

performance [14,20]. Finally, other studies evaluating the potential adverse impact of intrauterine hyperglycemia on offspring cognitive ability in O-mT1D have used O-BP as a reference group. However, to our knowledge, no previous studies have used O-fT1D as an unexposed reference group. We included O-fT1D as an unexposed reference group, thereby adjusting for any potentially shared genes for diabetes and impaired school performance. Also, having a parent suffering from a serious chronic disease like diabetes may in itself cause familial stress and be detrimental to school performance [23,24].

The present study has several limitations. First, the data did not include information on $HbA_{1c}$ during pregnancy because this information is not available in the Danish national registries. However, other Danish clinical studies on pregnant women with type 1 diabetes indicate that $HbA_{1c}$ is measured at least 5 times during pregnancy, and from 1992 to 1999, the $HbA_{1c}$ treatment target was not reached at any time during pregnancy according to the recommended $HbA_{1c}$ levels presently used in Denmark [3,25,26]. Also, during the years in which the included adolescents in our study were born, very few pregnant women with type 1 diabetes were using an insulin pump, and less than 50% were treated with human insulin [27,28]. Whether the glycemic level differed between mothers and fathers with type 1 diabetes during the time period covered by the present study can unfortunately not be explored using the information available in the Danish national registers.

Second, O-mT1D were more likely to have a missing test score than O-fT1D and O-BP. Nevertheless, sensitivity analyses on missing test scores did not change the conclusion.

Third, the current study only included children attending public schools because school tests are not mandatory in private schools. However, adjusted analyses revealed no significant difference between children attending public and those attending or private schools among the 3 groups.

Fourth, residual confounding cannot be excluded in studies based on a cohort design. Ethnicity or race was not considered in our study as this information is not available in Danish registers. However, immigrant or descendent status was included in our multivariable model, which to some extent counts as a proxy of ethnicity or race. Furthermore, obstetrics and perinatal complications are considered mediators on the casual pathway of maternal diabetes status and school performance; hence, adjusted analyses including these covariates are questionable because unknown confounding by factors affecting both the obstetrics and perinatal complications and school performance may introduce bias [29].

Finally, extrapolating our results to other populations should be done with caution because Denmark has a high-quality healthcare system that is free of charge, and all pregnant women with type 1 diabetes are referred for treatment at 4 highly specialized obstetric departments. Therefore, the difference in school performance may be even larger in countries where access to healthcare is seriously influenced by socioeconomic factors. Also, glycemic control in pregnant women with type 1 diabetes has improved in Denmark since the included cohort was born [30].

Parents or offspring with type 2 diabetes or mothers with gestational diabetes were not excluded from the analyses. However, we expect the potential effect of parents with either type 2 diabetes or gestational diabetes in O-BP or O-fT1D to be deluded by the large number of healthy parents in these groups, mainly because very few parents will have developed type 2 diabetes before birth due to their young age. Conceptually, we also think that the relevant comparison group is the "average child" in the background population, and by excluding children where the parents had other diagnoses than type 1 diabetes, the comparison group would be made healthier.

## Conclusions and further implications

Among Danish schoolchildren attending public school, O-mT1D achieved lower mean test scores than O-BP after adjusting for parental educational level and other confounding factors. However, mean test scores were indistinguishable from those in O-fT1D, indicating that the lower school performance in O-mT1D is associated with having a parent with a chronic disease rather than associated with intrauterine hyperglycemia. These results are reassuring for women with type 1 diabetes, as one of their main concerns is whether dysregulation of diabetes during their pregnancy may cause impaired cognitive development in their coming children. This study is, to our knowledge, the first to present evidence of an alternative explanation for the previously observed adverse effect of maternal type 1 diabetes during pregnancy on offspring cognitive development.

No direct information on $HbA_{1c}$ levels during pregnancy was available in our study; hence, we were not able to determine the association between $HbA_{1c}$ levels during pregnancy and school performance in O-mT1D, but this association is a potential subject for future studies in the field.

## Supporting information

**S1 Fig. Flow of study design.**
(TIF)

**S1 Text. Prospective analysis plan: School performance in Danish children exposed to maternal type 1 diabetes in utero: A nationwide retrospective cohort study.**
(DOCX)

**S1 Checklist. STROBE guideline.** STROBE, Strengthening the Reporting of Observational Studies in Epidemiology.
(DOCX)

**S1 Table. Multiple linear regression analyses comparing test scores in offspring of mothers (O-mT1D) and fathers (O-fT1D) with type 1 diabetes compared with offspring in the background population (O-BP), mean test score differencSSe (95% CI)—Model 3 with sequential adjustment for potential mediators.** O-BP, offspring of parents from the background population; O-fT1D, offspring of fathers with type 1 diabetes; O-mT1D, offspring of mothers with type 1 diabetes.
(DOCX)

## Author Contributions

**Conceptualization:** Anne Lærke Spangmose, Niels Skipper, Jannet Svensson, Tine Clausen.

**Data curation:** Anne Lærke Spangmose, Niels Skipper.

**Formal analysis:** Niels Skipper.

**Funding acquisition:** Anne Lærke Spangmose, Niels Skipper.

**Investigation:** Anne Lærke Spangmose, Niels Skipper, Jannet Svensson, Tine Clausen.

**Methodology:** Anne Lærke Spangmose, Niels Skipper, Jannet Svensson, Tine Clausen.

**Project administration:** Anne Lærke Spangmose, Niels Skipper, Jannet Svensson, Tine Clausen.

**Resources:** Niels Skipper.

**Supervision:** Sine Knorr, Tina Wullum Gundersen, Rikke Beck Jensen, Peter Damm, Erik Lykke Mortensen, Anja Pinborg, Jannet Svensson, Tine Clausen.

**Validation:** Anne Lærke Spangmose, Niels Skipper, Sine Knorr, Tina Wullum Gundersen, Rikke Beck Jensen, Peter Damm, Erik Lykke Mortensen, Anja Pinborg, Jannet Svensson, Tine Clausen.

**Visualization:** Niels Skipper.

**Writing – original draft:** Anne Lærke Spangmose, Niels Skipper.

**Writing – review & editing:** Sine Knorr, Tina Wullum Gundersen, Rikke Beck Jensen, Peter Damm, Erik Lykke Mortensen, Anja Pinborg, Jannet Svensson, Tine Clausen.

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
