## [Editor Report · Decision Letter 0]

3 Nov 2021

Dear Dr Spangmose, 

Thank you for submitting your manuscript entitled "School performance in Danish children exposed to maternal type 1 diabetes in utero: a nationwide cohort study" for consideration by PLOS Medicine.

Your manuscript has now been evaluated by the PLOS Medicine editorial staff and I am writing to let you know that we would like to send your submission out for external peer review.

Kind regards,

Beryne Odeny

Senior Editor

PLOS Medicine

---

## [Decision Letter · Decision Letter 1]

20 Jan 2022

Dear Dr. Spangmose,

Thank you very much for submitting your manuscript "School performance in Danish children exposed to maternal type 1 diabetes in utero: a nationwide cohort study" (PMEDICINE-D-21-04566R1) for consideration at PLOS Medicine. 

[LINK]

In light of these reviews, I am afraid that we will not be able to accept the manuscript for publication in the journal in its current form, but we would like to consider a revised version that addresses the reviewers' and editors' comments. Obviously we cannot make any decision about publication until we have seen the revised manuscript and your response, and we plan to seek re-review by one or more of the reviewers. 

We expect to receive your revised manuscript by Feb 10 2022 11:59PM. Please email us (plosmedicine@plos.org) if you have any questions or concerns.

We look forward to receiving your revised manuscript. 

Sincerely,

Beryne Odeny, 

PLOS Medicine

plosmedicine.org

1) Please revise your title according to PLOS Medicine's style. Please consider “School performance in Danish children exposed to maternal type 1 diabetes in utero: a nationwide retrospective cohort study”

2) The Data Availability Statement (DAS) requires revision. For each data source used in your study, if the data are not freely available, please describe briefly the ethical, legal, or contractual restriction that prevents you from sharing it. Please also include an appropriate contact (web or email address) for inquiries (this cannot be a study author).

3) Please move the ethical approval statement from the end of the main text to the Methods section.

4) Abstract:

a) Please combine the Methods and Findings sections into one section, “Methods and findings”

b) Please ensure that all numbers presented in the abstract are present and identical to numbers presented in the main manuscript text.

c) Please quantify the main results (with 95% CIs and p values).

d) In the last sentence of the Abstract Methods and Findings section, please describe the main limitation(s) of the study's methodology.

e) Conclusions: Please address the study implications without overreaching what can be concluded from the data and emphasize what is new considering the current evidence.

f) Please remove the funding statement from the end of the abstract

6) Please avoid assertions of primacy ("… to provide among/ some of the first evidence...."). Instead, state “to our knowledge” or similar.

7) Please conclude the Introduction with a clear description of the study hypothesis.

8) Did your study have a prospective protocol or analysis plan? Please state this (either way) early in the Methods section. 

9) Please state whether the health data were de-identified (anonymized) prior to access.

10) Please ensure that the study is reported according to the STROBE guideline, and include the completed STROBE checklist as Supporting Information. Please add the following statement, or similar, to the Methods: "This study is reported as per the Strengthening the Reporting of Observational Studies in Epidemiology (STROBE) guideline (S1 Checklist)."

11) When completing the STROBE checklist, please use section and paragraph numbers, rather than page numbers.

12) Please avoid assertions of primacy (“This is the first study to...."). Instead, state “to our knowledge” or similar.

13) Was ethnicity or race considered in this study? 

14) In your statistical analyses, please account for clustering of data at the school level. Generalized Estimating Equations (GEE) or hierarchical/ multilevel models, among others, may be useful in this case.

15) We note the potential for unobservable confounding in this observational study. Have you considered using robust methods such as propensity score matching to address this?

16) Please discuss whether missing scores were missing (at random/ completely at random) and the justification for your approach to handling missing data.

17) In the results text and tables, please quantify the main results with both 95% CIs and p values (where applicable).

18) Throughout the text, please remove language that implies causality, such as “more likely explained by” in the discussion and conclusion sections. Refer to associations instead.

19) References:

a) Please select the PLOS Medicine reference style in your citation manager. In-text reference call outs should be presented as follows noting the absence of spaces within the square brackets: "... countries [1,2]."

b) In the reference section, please ensure six names appear before et al.

20) Please remove the “Data sharing”, “Declaration of Interests”, “transparency”, and “Funding” statements at the end of the main text. This information is captured in the metadata obtained in the submission form.

Comments from the reviewers:

Reviewer #1: "School performance in Danish children exposed to maternal type 1 diabetes in utero: a nationwide cohort study" studies both the possible adverse effects of mothers with Type 1 diabetes (O-mT1D, n=2,144) and fathers with Type 1 diabetes (O-fT1D, n=3,474) on their offspring, against a background population (O-BP, n=616,455). This was performed as a population-based retrospective cohort study with data from 2010 to 2016 using nationally-standardized test scores for reading and math, at various public school grade levels. Adjusted (pooled) analyses found statistically-significant (adverse) mean test score difference of -1.62 for o-mT1D and −0.86 for O-fT1D, against the control group. Various sensitivity analyses involving different sets of covariates were attempted (Table 2, Supplement Table 1), with generally consistent outcomes for the T1D vs. control groups.

The scale of the study and the examination of fathers with Type 1 diabetes are particular strengths, with the latter relatively novel to the area, and relevant to answering hypotheses about potential mechanisms for the observed adverse effects relating to non-maternal factors. Some issues might however be considered:

1. Inclusion/exclusion and group criteria might be considered to be summarized in a flowchart if possible.

2. Related to the above, the treatment of other diabetes types (Type 2, gestational, etc.) in both parents and offspring might be clarified, prefably also in the flowchart if included. Were they excluded/covariates?

3. In Line 99, it is stated that "Offspring in whom both the mother and the father were diagnosed with diabetes (n=30, Line 192) were excluded". It might be clarified if this refers to T1D only for both, and the number of such cases.

4. In Line 144, it is stated that test scores were analyzed in a pooled analysis, with each test score (assumed for different topics/grades) as a unique observation. It might be discussed as to whether pooling across topics/time is appropriate, for example since reading scores would seem to be represented twice as much as math scores, simply by (arbitrary) virtue of having twice as many tests taken. More details on how the pooling was setup might be provided, possibly in supplementary material. Nonetheless, individual tests appear to generally support the main findings (Figure 1).

5. While some prior work has been cited as justification for the choice of covariates (Line 152), certain common covariates (e.g. age at each test, ethnicity, individual school performance) oft considered in studies involving student test scores, appear not to have been included. It might be briefly clarified as to whether such effects are assumed to have been covered by other included covariates.

6. In Line 207, it is stated that "The mean of test scores in both math and reading were 54.2 (SD 24.8) in O-mT1D, 54.4 (SD 24.8) in O-fT1D, and 56.4 (SD 24.7) in O-BP". It is assumed that the reported figures are the means for (math and reading) together, and not that the individual means (and SD) were identical; as such, the phrasing might be considered to be reworded (including whether the weighing for math and reading were identical), and results for individual topics might be considered for discussion.

Minor issues:

7. In Line 88, "inclusion of only offspring born in Denmark in whom there was..." might be considered to be reworded to avoid the phrase "only offspring", which might suggest children without siblings, if that is not the intended meaning.

Reviewer #2: School performance in Danish children exposed to maternal type 1 diabetes in utero: a nationwide cohort study

Overall:

An interesting twist to examination of the effect of in utero exposure to type 1 diabetes is described by using a comparison group of fathers with type 1 diabetes rather than just those offspring without exposure to type 1 diabetes in utero. The linking of databases in Denmark provides a unique opportunity to examine the outcomes of interest.

Children with both parents having type 1 diabetes were excluded, n=30.

From what I can tell rates of diabetes were increasing in the female population of reproductive age during the gestation of the offspring studied, mostly type 1 but there could have been an increase in type 2 and gestational diabetes. My question is, were these offspring, from women with type 2 or gestational diabetes during the pregnancy, removed from the comparison groups?

The introduction brings up the point that there are few human studies that examine the type 1, type 2 and gestational diabetes separately and these are only small studies. I would have thought the authors may have been able to distinguish between the type of in utero diabetes exposure in this population using ICD codes. These fetus' are exposed to hyper- and hypo-glycemia also. I feel that these offspring do not belong in the comparison groups. However, given the sample size this may be a moot point, but at least worth mentioning in the limitations.

Specific questions:

Lines 116-117: I am not sure of the importance of the likelihood of attending a private school as an outcome. I see the use of the likelihood of missing a test in the public school but am not clear about the former. Reporting of the results lines181 to 183 clarifies this. Please make it clear in the methods the reasoning for this "outcome", although I believe it is more a matter of checking that the population under study is representative.

Lines 190-192: It would be useful to the reader to repeat the number of offspring.

"2144 offspring of 1597 unique mothers and 3474 offspring of 2419 fathers with type 1 diabetes, and 616 455 offspring of 401 406 mothers and 402 785 fathers from the background population were included in the cohort."

Of note - I don't believe they were all adolescents unless I am misunderstanding the population. Mean age at first test was 10.1, but as many had more than one test, I am not sure what age is representative of the population. It might be useful to put age at first test for each group in table 1 under the "Test" as age at test is included in the models.

I am surprised by inclusion of both maternal and paternal education and income in model 3, I am presuming collinearity was checked. Sorry, I am confused, in re-reading the notes I see that parental highest education level and income were included in the model. Is that highest income or total, is it a category of percentile rank or a continuous variable? Please clarify. Maybe it would be preferable to only show in the table the variables included in the model.

The differences between math and reading results are fascinating and I feel that something is lost by putting both in the same model, there also appear to be grade differences. Model 3 is also used for these analyses, were other models considered?

Reviewer #3: 

Thank you for the opportunity to review this paper.

Utilising national population data, the authors primarily examined the association between maternal typeI diabetes on offspring school performance, but also, in respect to fathers with type1 diabetes and the background population. This is an excellent study and the paper is well written and succinct. Inclusion of fathers with diabetes is particularly important and adds significantly to our understanding of developmental origins of human capital.

Minor points for consideration:

1. The inclusion of both parity and number of siblings may over-adjust the model - have the authors examined collinearity effects and is there strong enough justification to include both variables?

2. As anticipated, sex does have an association with school performance and is appropriate to adjust for. However, there are data suggestion a sexual dimorphism in the effect of maternal glucose on fatal growth, particularly for males. Given the sufficient power of the sample size, it would useful to sex-stratify and examine the association of maternal and paternal diabetes on the math outcome in sex-specific models to test if diabetes exposure effects male school performance more.

3. Figure 1 is useful to include but the view quality of the figure needs to be improved

A potential confounder is how well maternal glucose was managed during pregnancy - it is unfortunate that no data could be presented on this, but the authors have discussed this within the limitation section.

[LINK]

---

## [Decision Letter · Decision Letter 2]

16 Mar 2022

Dear Dr. Spangmose,

Thank you very much for re-submitting your manuscript "School performance in Danish children exposed to maternal type 1 diabetes in utero: a nationwide retrospective cohort study" (PMEDICINE-D-21-04566R2) for review by PLOS Medicine.

I have discussed the paper with my colleagues and the academic editor and it was also seen again by one reviewer. I am pleased to say that provided the remaining editorial and production issues are dealt with we are planning to accept the paper for publication in the journal.

Please address the remaining issues within 1 week to enable publication of your manuscript in April. The remaining issues that need to be addressed are listed at the end of this email. Any accompanying reviewer attachments can be seen via the link below. Please take these into account before resubmitting your manuscript:

[LINK]

We look forward to receiving the revised manuscript by Mar 23 2022 11:59PM.   

Sincerely,

Beryne Odeny, 

PLOS Medicine

plosmedicine.org

1) Abstract conclusion: please add a sentence addressing the study implications for policy and practice.

2) Thank you for providing your STROBE checklist. Please replace the page numbers with paragraph numbers per section (e.g. "Methods, paragraph 1"), since the page numbers of the final published paper may be different from the page numbers in the current manuscript.

3) In table 2, please define what the income percentile ranks mean e.g. Q1 is which income range etc

4) To help us extend the reach of your research, please provide any Twitter handle(s) that would be appropriate to tag, including your own, your coauthors’, your institution, funder, or lab.

Requests from Editors:

Comments from Reviewers:

Reviewer #1: We thank the authors for addressing our previous comments. A few further points might be considered:

1. The clarifications for other diabetes types (involving Table R1 in the response) might be included in supplementary material, and/or briefly mentioned in the text, if possible.

2. The test score pooling setup might be described in greater detail, if possible (and offered).

3. Limitations relating to available covariates might be briefly discussed.

4. Results as presented in Table 2 appear changed from the previous revision, despite apparently involving the same population. This might be clarified.

[LINK]

---

## [Editor Report · Decision Letter 3]

29 Mar 2022

Dear Dr Spangmose, 

On behalf of my colleagues and the Academic Editor, Dr. Lars Åke Persson, I am pleased to inform you that we have agreed to publish your manuscript "School performance in Danish children exposed to maternal type 1 diabetes in utero: a nationwide retrospective cohort study" (PMEDICINE-D-21-04566R3) in PLOS Medicine.

PRESS

Sincerely, 

Beryne Odeny 

PLOS Medicine